# Basic Knowledge and New Advances in Panoramic Radiography Imaging Techniques: A Narrative Review on What Dentists and Radiologists Should Know

Rossana Izzetti [1], Marco Nisi [1], Giacomo Aringhieri [2], Laura Crocetti [2], Filippo Graziani [1,*] and Cosimo Nardi [3]

1   Unit of Dentistry and Oral Surgery, Department of Surgical, Medical and Molecular Pathology and Critical Care Medicine, University of Pisa, 56126 Pisa, Italy; ross.izzetti@gmail.com (R.I.); marco.nisi@unipi.it (M.N.)
2   Diagnostic and Interventional Radiology, Department of Translational Research and of New Technologies in Medicine and Surgery, University of Pisa, 56122 Pisa, Italy; giacomo.aringhieri@unipi.it (G.A.); laura.crocetti@med.unipi.it (L.C.)
3   Radiodiagnostic Unit n. 2, Department of Experimental and Clinical Biomedical Sciences, University of Florence—Azienda Ospedaliero-Universitaria Careggi, Largo Brambilla 3, 50134 Florence, Italy; cosimo.nardi@unifi.it
*   Correspondence: filippo.graziani@med.unipi.it

**Abstract:** Objectives: A panoramic radiograph (PAN) is the most frequently diagnostic imaging technique carried out in dentistry and oral surgery. The correct performance of image acquisition is crucial to obtain adequate image quality. The aim of the present study is to (i) review the principles of PAN image acquisition and (ii) describe positioning errors and artefacts that may affect PAN image quality. Methods: Articles regarding PAN acquisition principles, patient's positioning errors, artefacts, and image quality were retrieved from the literature. Results: Head orientation is of the utmost importance in guaranteeing correct image acquisition. Symmetry, occlusal plane inclination, mandibular condyles localization, cervical spine position, aspect of upper teeth root apexes, exposure parameters, and metal and motion artefacts are factors that greatly affect the image quality of a successful PAN. Conclusions: Several factors are the basis for PAN performance; therefore, a systematic approach that takes into account correct patient positioning and preparation is strongly suggested to improve overall examination quality.

**Keywords:** panoramic radiography; orthopantomography; errors; artefacts; diagnostic imaging; image quality

## 1. Introduction

Panoramic radiography (orthopantomography, PAN) is the most frequently prescribed screening examination in dentistry [1,2]. This technique has several advantages, particularly a relatively low cost, a low radiation dose, and the possibility of obtaining a comprehensive overview of dental arches, maxillary and mandibular bones, as well as of relevant anatomic structures to be preserved during surgery, such as inferior alveolar nerves and maxillary sinuses [3–5]. Moreover, PAN has been found to have a high number of line pairs per millimeter (lp/mm), ranging from 1.6 to 3.0 lp/mm [6,7]. Interestingly, according to the literature, PAN has a better number of line pairs than cone beam computed tomography, which is reported to range between 0.6 and 2.8 lp/mm in experimental settings and decreases down to above 1 lp/mm in clinical settings [8].

However, PAN only provides a two-dimensional representation of a three-dimensional area. The superimposition of different structures may hinder the correct interpretation of examinations and can sometimes be misleading [9,10]. Moreover, the image deformation deriving from the geometry of acquisitions can result in erroneous measurements and evaluations. Finally, the lack of fine details of PAN compared to volumetric imaging

techniques such as cone beam computed tomography can lead to the misinterpretations of possible alterations [11].

Along with the aforementioned issues related to the performance of PAN, an additional problem is represented by errors in patient positioning, which result in a decrease in the diagnostic quality of examinations. The correct positioning of a patient is crucial to attain adequate image quality that focuses on the teeth and alveolar bone structures. Positioning errors have l been poorly discussed and need to be understood by radiologists and dentists to avoid unnecessary radiation re-exposure [12–21].

The aim of the present study is to review the principles of PAN image acquisition and the most common errors that occur during its execution to provide clinicians a guide for the correct performance of this imaging technique.

## 2. PAN Acquisition Principles

PAN is a zonography performed by means of a simultaneous rotation of an X-ray beam generator and a detector system—curved rotational thick-layer tomography—that are both mounted on a rotating gantry and positioned on either side of patient [10,22–24]. The X-ray tube rotates clockwise around the patient's head from right to left, passing behind the shoulders and making an arc of about 270°. The final image is the result of reconstructions of individual image sections of the maxillofacial area [10,20–25]. The one-to-one correspondence between each point of dental arches crossed by a collimated beam and its projection on the detector is the basis of PAN image formation.

During an examination, a patient can be in a standing or sitting position. The patient's head is kept in position through a chin, a forehead rest, and lateral head supports. Indicator lights guide patient positioning [23,24]. The midsagittal plane should be centered in the rotational midline of the device and should be perpendicularly aligned to the floor. The Frankfort plane, that is the line passing from the superior border of the external auditory meatus to the infraorbital rim, should be horizontally oriented and thus parallel to the floor [12–14]. A notched bite block guides the position of the maxillary and mandibular incisors. Head-to-head positioning of the incisors of both arches on the same plane within the focal trough coincides with the rotation fulcrum—center of symmetry—of a panoramic dental unit to assure the passage of radiation right through these teeth. Bite blocks also provide distancing between teeth to avoid crown superimposition [16,17]. During the examination, the spine has to be as straight as possible, and the neck should be extended. This position may be obtained through a slight lower inclination of chin, with the feet parallel and anteriorly related to body [26,27]. This because the cervical spine has to remain within the focal trough with no superimposition on the anterior teeth. In addition, a straight spine and an elongated neck reduce the possibility of the X-ray tube bumping into the back or shoulders of patients with a big physical stature or obvious postural thoracic kyphosis [12–14]. Furthermore, patients have to keep the tongue pressed against the palate (roof of the mouth) to avoid both overexposure and the presence of a radiolucent area representing the shadow of the air space on the upper teeth apexes [23–25,27].

## 3. The Variables of a Correct PAN Examination

PAN is characterized by the geometric projection of structures located in the maxillofacial area. The acquisition modality accounts for geometric distortions occurring to the anatomical structures depending on their position with respect to a radiation source [14,16,17]. In particular, structures located closer to the source appear increased in their dimensions; conversely, structures located far from the source appear smaller [26]. The shape and width of the focal trough are determined by the path and velocity of the detector and the X-ray tube head, alignment of the X-ray beam, and collimator width [26,27]. Generally, the focal trough—also called focal plane or image layer—in PAN has limited size, ranging from 6 mm to 8 mm and 12 mm to 16 mm in anterior and posterior areas, respectively. Distinct imaging of the structures located within the focal trough is obtained, with the structures positioned in the center of the focal trough being the clearest and the

objects located outside the focal trough appearing blurred, distorted, or reduced/magnified. Precisely, teeth and other structures appear blurred, shortened, and narrowed, or blurred and expanded in the horizontal direction when such structures are placed in front of or behind the focal trough, respectively [14]. Geometric distortions of anatomic structures are minimal in the intercanine area, since the anterior area is included in the focal trough by means of an obligated allocation of the incisors in bite blocks. In molar areas, on the contrary, geometric distortions are more noticeable (5–10% compared to actual size) because of interindividual anatomic variability in dental arch conformations [13,14].

## 4. Frequency of Errors in PAN Performance and Interpretation

The frequency of positioning errors when performing PAN has seldom been described in the literature. In a recent review [28], it was reported that PAN have a mean reject rate of 4% due to positioning errors and motion artefacts. On average, one to three errors per PAN can be found in examinations of permanent dentition, including wrong tongue positioning, lack of stabilization of the chin on the chin rest, and a retruded position with respect to the focal trough [17]. According to Rondon et al. [16], patient head positioning accounted for the majority of cases that required re-examination. In particular, incorrect head positioning related to the focal trough, lateral rotation, and forward inclination represented the most common errors, and these were each encountered in more than 20% of cases. In a study [20] on 200 PAN, only 36% of cases did not show positioning errors. The presence of radiolucency obscuring the roots of maxillary teeth was observed in 33% of cases, followed by artifacts, reflected and ghost images (17%), backward head inclination (15%), spine overlapping in the mandibular anterior region (10%), and teeth positioned behind or in front of the focal trough (7% and 1%, respectively). Similarly, Loughlin et al. [29] performed a quality assessment of 315 PANs taken either at general radiology departments or in dental radiology units. They found that only 20% of these PANs could be rated as excellent in the absence of errors in patient preparation, exposure, positioning, and image processing. However, 70% of the examinations were considered diagnostically acceptable, since the errors did not hinder diagnostic performance, whereas only 10% were considered unacceptable. Among the most frequently occurring errors, incorrect patient preparation (especially wrong tongue positioning, overlapping of upper and lower incisors, and the wearing of removable appliances/jewelry) and head positioning (rotation, upper chin position) were the most commonly found. Conversely, motion artefacts and exposure errors were infrequently reported. A randomized, controlled trial by Wenzel et al. [30] tested the ability of dental students to perform a correct PAN following a computer-assisted learning program, simulation training, or conventional teaching. Errors in the positioning of the horizontal plane occurred in all the three groups. Rotation and positioning errors on the sagittal plane were the most rarely encountered, whereas correct coronal plane positioning was only achieved in the simulation training group. The authors concluded that training significantly improved skills in relation to performing PAN.

## 5. The Importance of Head Orientation

Head positioning plays a major role in the correct performance of a PAN examination. Errors in head positioning can occur on horizontal, vertical, and anteroposterior planes, or may result from a combination of multiple (compound) errors in more than one plane [14–16,24–26]. To simplify our analysis of head positioning, horizontal and vertical head movements are defined as tilting and nodding, respectively, whereas rotational head movements are defined as rolling (Figure 1) [31,32].

Errors in horizontal positioning can occur when a patient's head is tilted (tilting) or twisted (rolling), causing a wrong alignment on the midsagittal plane. Accordingly, anatomic structures appear asymmetrically enlarged on one side and reduced on the opposite side [14–16]. To avoid errors in head positioning on the horizontal plane, the midsagittal plane should be placed perpendicular to the floor and correspond to patient's midline.

The lateral head supports can help a patient's stabilization and prevent tilting/rolling movement during the exposure [24,25].

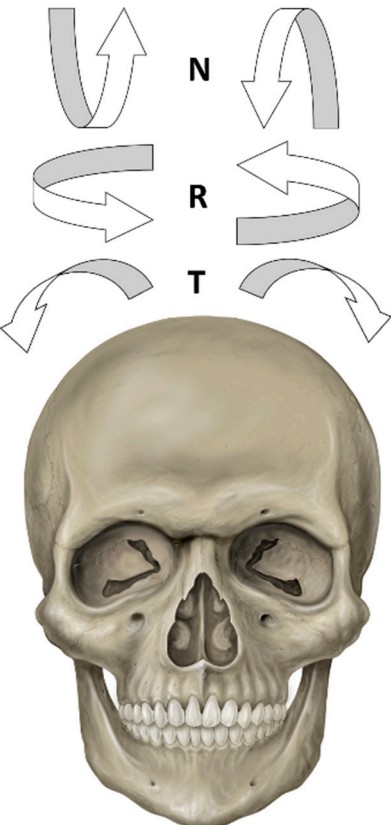

**Figure 1.** Types of movement of patient's head. N: nodding. R: rolling. T: tilting.

In case of rolling, the mandibular ramus width and teeth size decrease on the opposite side to the head's movement, since the opposite side in which the head is twisted is closer to the detector. When the head is twisted to one side, both mandibular ramus and teeth on the opposite side appear shortened (Figure 2) [14–16].

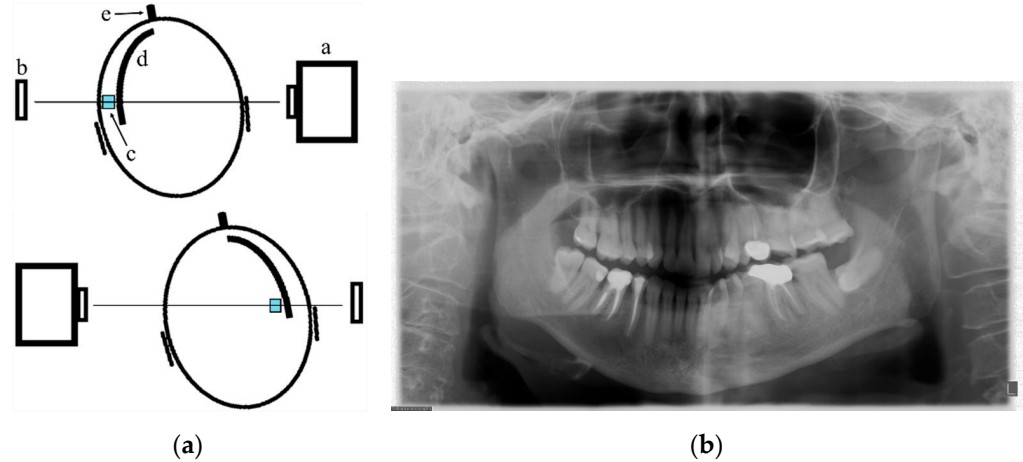

(**a**)　　　　　　　　　　　　　　　　　　　　　　　　(**b**)

**Figure 2.** Rolling. Head rotation towards the left: (**a**) schematic drawing. Posterior teeth move to the right and do not coincide with the focal trough. Left hemi-arch is close to the X-ray radiation tube with consequent magnification of left hemi-arch teeth and mandibular ramus. On the contrary, right

hemi-arch is close to the detector with consequent contraction in size of right hemi-arch teeth and mandibular ramus. a: X-ray radiation tube. b: detector. c: focal trough. d: dental arches. e: nose. (**b**) Panoramic radiography. Note the vertical dark band in the paraincisal area on the right side corresponding to the asymmetric position of two strong X-ray-absorbing structures, i.e., cervical spine and occipital bone. Premolar and molar teeth are blurry and increase in size on the left side; in addition, an overlapping of premolar teeth can be observed.

In case of tilting, both the occlusal plane and mandibular lower border are inclined towards the tilted side. When the head is tilted to the left, the occiput is closer to the right side, causing an enlargement of the left side due to its closer position to the radiation source and a farther one from the detector. Conversely, the right side will appear reduced in size, as it is farther from the radiation source and closer to the detector (Figure 3) [12–14].

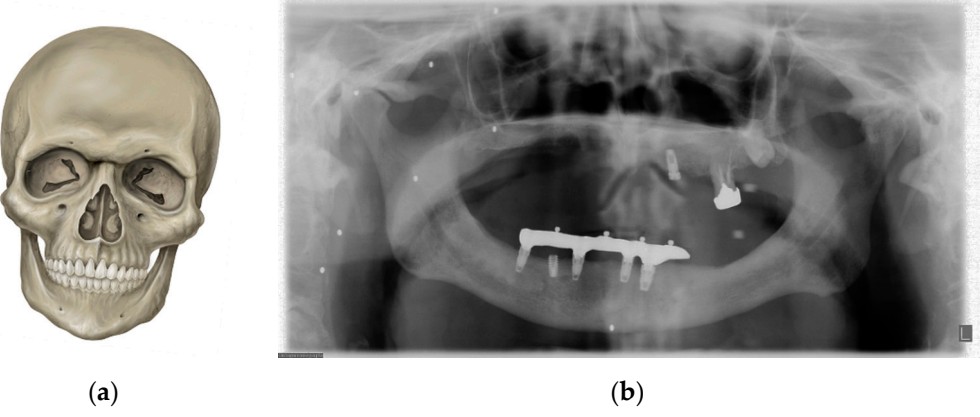

(**a**)                                                                                                        (**b**)

**Figure 3.** Tilting. Head tilt towards the left: (**a**) picture; (**b**) panoramic radiography. The left mandibular angle and condyle are in a lower position than the contralateral side. Note the punctiform metallic images on the right side due to the shotgun pellets in the soft tissue. Metal artefacts are projected into the airspace between upper and lower jaws on the left side in the form of punctiform radiopaque images.

Errors in vertical positioning (nodding) affect the inclination of the occlusal plane. Vertical errors are related to an incorrect upper or lower position of the chin. If the chin is positioned higher, the hard palate is superimposed over the maxillary teeth apexes and condyles are dislocated posteriorly, or even projected off the image because of the increasing intercondylar distance. The nasal cavity and maxillary sinuses appear blurred and out of focus, and upper teeth are elongated (Figure 4) [12–14]. Conversely, if the head is nodded down, the upper and lower jaws appear magnified. In addition, the anterior mandibular area is partially out of the focal trough and may be superimposed on the hyoid bone (Figure 5) [14–16].

Errors on the anteroposterior plane are often related to an incorrect positioning of a patient's teeth on the bite block, which may result in excessive forward or backward positioning. If a patient's position is too anterior, the anterior teeth of both arches are placed out of the focal trough, and thus appear narrowed and blurred; however, the spine, nasal fossa, and maxillary sinuses are included in the focal trough. The cervical spine shows superimposition over mandibular ramus on both sides. Conversely, a backward position causes the ghosting of mandibular ramus, an increased size of the anterior upper and lower teeth, and an overall widening of anatomic structures, cutting off posterior ones (Figure 6) [14–16,25].

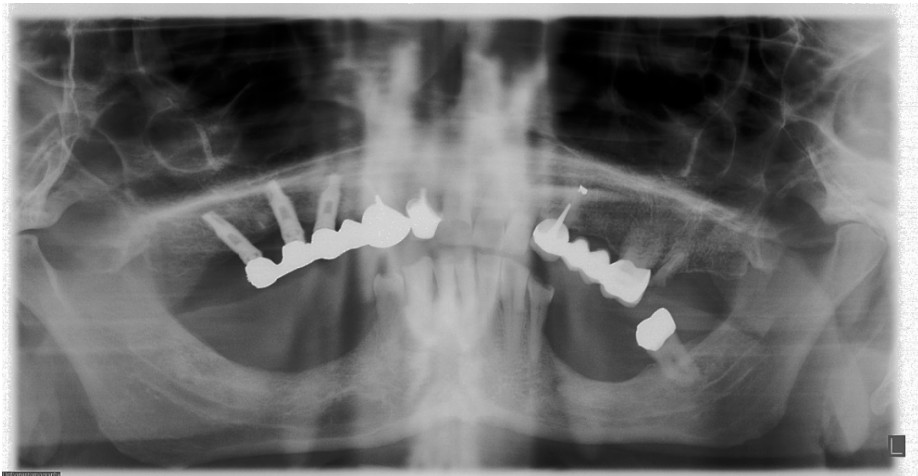

**Figure 4.** Nodding up. Panoramic radiography. The occlusal plane loses the normal upper concavity and its curve is inverted. Note the radiopaque band overlapping the anterior area of both jaws.

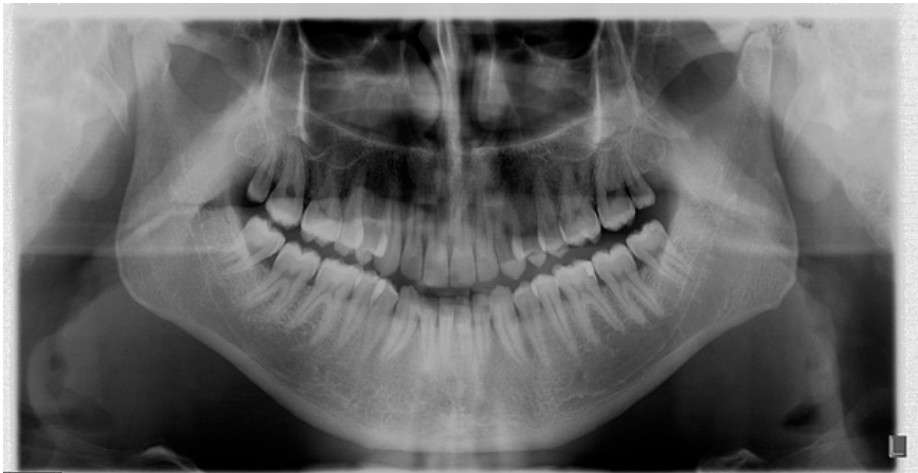

**Figure 5.** Nodding down. Panoramic radiography. The upper concavity of the occlusal plane is pronounced.

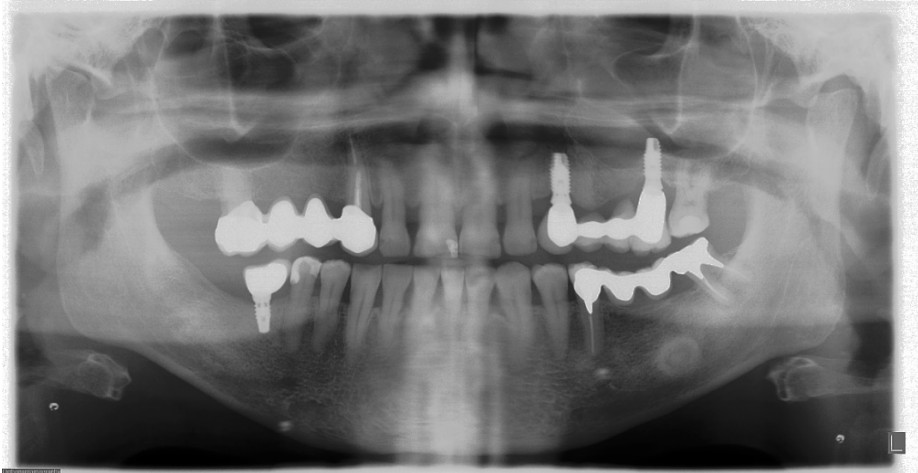

**Figure 6.** Backward position. Panoramic radiography. Anterior teeth of both arches are placed behind the focal trough, with a consequent increase in their size (incisive teeth seem larger than molar ones). Note the radiopaque band overlapping the mandibular incisive area.

## 6. Evaluation of PAN Examination

A correct PAN requires the cautious positioning of a patient and an appropriate technique. Overall, six parameters define whether a PAN is correctly performed, namely (i) symmetry of the two sides, (ii) occlusal plane with a mild upper concavity, (iii) localization of the two mandibular condyles at the same height, (iv) clear representation of dental apexes of upper teeth, (v) straight position of the cervical spine, and (vi) adequate exposure parameters [26,27].

### 6.1. Symmetry

Horizontal plane positioning is crucial for obtaining PAN image symmetry. When head tilting and rolling occur, an asymmetry in the PAN can be observed. In these cases, one side appears increased in size compared to the other one, which means that the magnified area is placed closer to the radiation source and can thus be related to a lateral orientation of patient's head. If errors in horizontal positioning are present, an asymmetric vertical dark band can be observed in the paraincisal area; this corresponds to the asymmetric position of two strong X-ray-absorbing structures, i.e., the cervical spine and occipital bone [26,27,31]. In cases where the positioning of the midsagittal plane has failed, an uneven magnification in the horizontal size of right and left sides can be observed, along with horizontal distortion in posterior areas and excessive tooth overlapping in premolar areas. The side located closer to the detector appears reduced in size, whereas the side positioned closer to the rotation center appears enlarged (Figure 2) [16,17].

### 6.2. Inclination of the Occlusal Plane

The inclination of the occlusal plane is strictly related to the position of the Frankfort plane, since the two planes tend to converge posteriorly [24,25]. When a PAN is correctly performed, the X-ray beam is parallel to the Frankfort plane, which causes the posterior areas to be placed in a higher position than the anterior ones (Figure 7) [15]. If a patient's head is nodded down—teeth anterior to the focal trough—the concavity of the occlusal plane appears increased (Figure 5). Conversely, an upwards movement of the chin with a retroflex head—teeth posterior to the focal trough—results in a flattening or inversion of the occlusal plane curve, with a distortion of the mandible and a superimposition of the hard palate on maxillary teeth roots (Figure 4) [18,33].

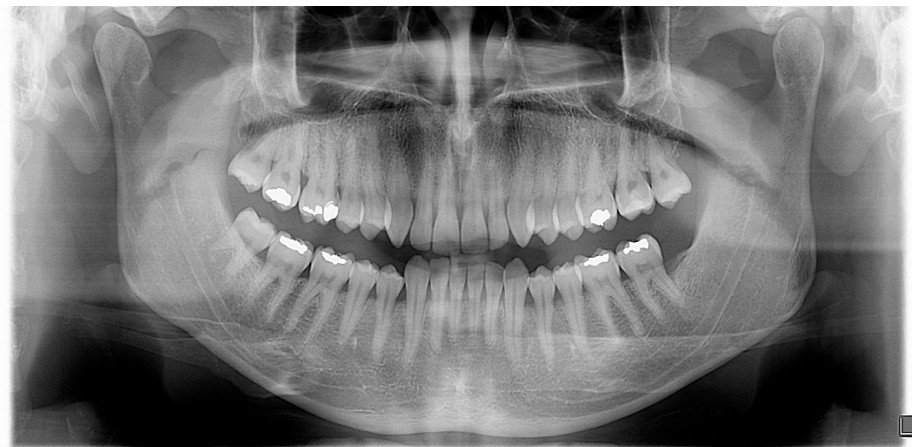

**Figure 7.** Panoramic radiography carried out perfectly. The dark space between the two dental arches represented by the occlusal plane shows a mild upper concavity, since the Frankfort and occlusal planes tend to converge posteriorly.

### 6.3. Localization of Mandibular Condyles

The position of mandibular condyles is strictly related to the correct orientation of a patient's head on the horizontal plane. If the patient's head is tilted, the mandibular angles

are positioned one higher than the other; condyles are also at different heights [20,21]. In case of rolling, the mandibular condyle located on the same side in which the head is twisted appears enlarged, since it is closer to the radiation source (Figure 3) [14,15].

### 6.4. Aspect of Upper Teeth Root Apexes

The correct visualization of all upper teeth root apexes is obtained when patients are instructed to swallow and hold the tongue flat against the hard palate in order to eliminate the dark airspace between two such anatomic structures [26,27]. Therefore, when the dorsum of the tongue does not perfectly adhere to the roof of the mouth, a radiolucent band at the apex of upper teeth obscures the maxillary periapical bone tissues. Moreover, crowns of incisors can be obscured by the air space if the lips remain open (Figure 8) [1–14].

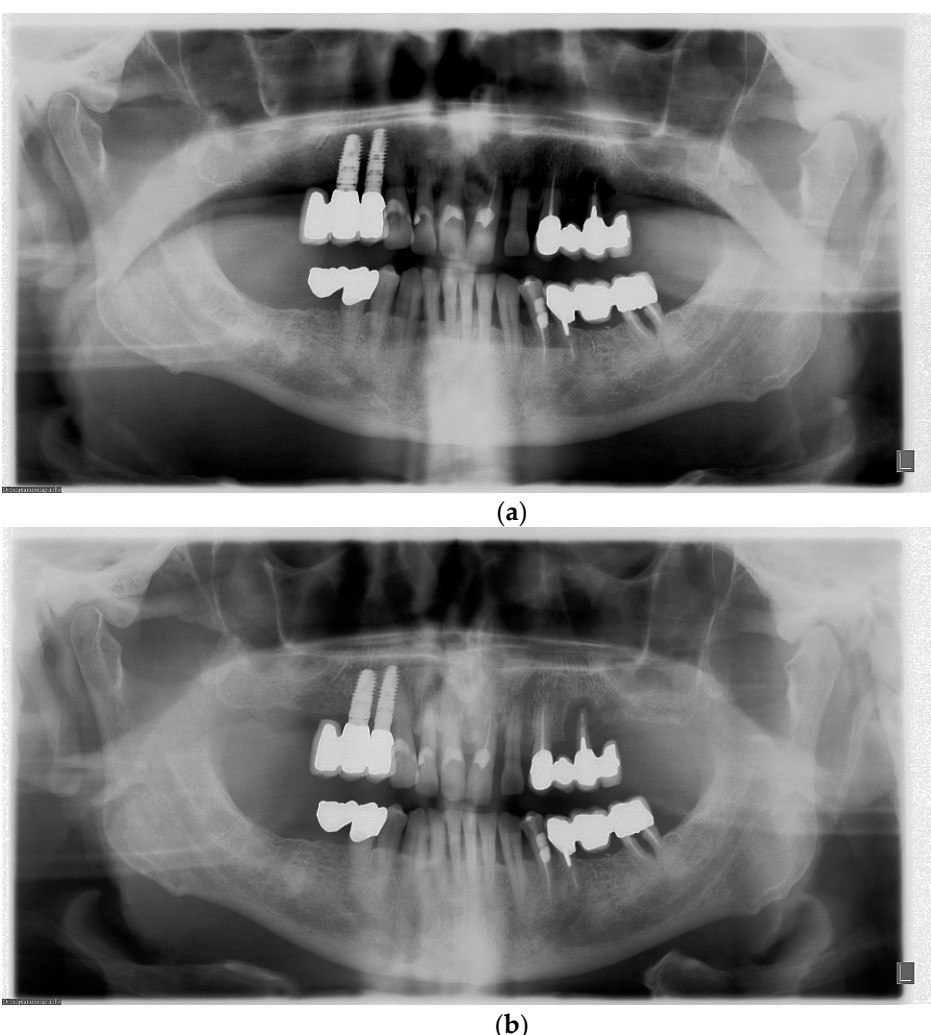

(a)

(b)

**Figure 8.** Position of the tongue: (**a**) tongue does not press against the palate. The shadow of the airspace causes a radiolucent area covering the upper teeth. (**b**) Tongue presses against the palate. The maxillary periapical bone tissues can be appropriately identified since the tongue acts as filter effect by absorbing X-rays in correspondence with the upper teeth roots. Note the radiopaque band overlapping the incisive area of both jaws.

### 6.5. Position of the Cervical Spine

The cervical spine has to be located inside the focal trough. When the cervical spine is not correctly positioned, this may hinder the visualization of the anterior region, and it may appear as a vertical radiopaque shadow obscuring the mandibular symphyseal area and, to a lesser extent, the anterior maxillary area [14–16,18] (Figures 4, 6 and 8). A

rachis superimposition has to be limited by positioning patients according to the so-called water-skier position that requires a patient to be in a standing position with an extended neck, their shoulders down, a straight back, and their feet together in an anterior position compared to the torso [16–18].

*6.6. Exposure Parameters and Radiation Dose*

There are three main factors that can affect the quality of a PAN examination, namely kilovoltage setting, incorrect program selection, and incomplete acquisition [33–35].

Errors in tube voltage or current settings are the most common errors. Generally, PAN requires tube voltages of 70–80 kV and current intensity ranges from 8 to 12 mA. Overexposed images are characterized by a blackening of PAN due to an excessively high tube voltage or current intensity, along with prolonged exposure time. Conversely, low kilovoltage or current intensity and insufficient exposure time produce underexposed images [25]. Digital panoramic dental units enable automatic modulations of X-ray beam energy, ensuring an optimal exposure control for maximizing image quality and limiting both technical errors of over/underexposure and radiation dose. Digital radiography image detectors have a higher dynamic range than conventional analogue films, which means that digital image detectors respond to X-ray beams and produce data in a wider range of X-ray exposure values [27]. In cases of overexposure, image post-processing with contrast and/or brightness corrections can help to increase the quality of a digital image which, in an analogue variety, would have been diagnostically unacceptable. On the contrary, underexposed images often do not show significant improvement by image post-processing; therefore, in some cases, images need to be retaken [33,34]. The effective dose for conventional analogue PAN ranges from 15 to 40 μSv depending on the machine, whereas it is around half for digital PAN when the lowest possible radiographic settings are used [11,36]. The advantages and disadvantages of digital radiography compared to analogue radiography are summarized in Table 1.

**Table 1.** Advantages and disadvantages of digital panoramic radiography.

| Digital Panoramic Radiography | |
| --- | --- |
| *Advantages* | *Disadvantages* |
| Ease of execution | One-time high cost of implementation of the digital system |
| Significant reduction in time between exposure and ready radiograph | Need for knowledge of computing technology by operators |
| Image post-processing options | Lack of control over radiographs with too many retaking examinations |
| Long-lasting archive of images with the easy retrieval of multiple copies | Need to protect sensitive patient data |
| Teleradiology. Image transfer and reporting via internet | Possibility of tampering with images |
| Wide dynamic range of X-ray exposures | |
| Automatic modulation of X-ray beam energy | |
| Decrease in radiation dose | |

During the movement of the X-ray tube detection system, rotation velocity decreases when the X-ray tube is located behind a patient's head due to the presence of bone structures that absorb X-rays, such as the cervical spine and occiput. This feature ensures the correct exposure of the anterior area. However, even if tube voltage, current intensity, and exposure time are perfectly selected, an insufficient coverage of structures of interest can be found when the selected program in a panoramic dental unit is incorrect [33–35].

Finally, a partial image is obtained in cases of early release of the exposure button prior to the completion of the entire exposure cycle [25,35]. At a time when the X-ray tube stops, a sharp image of the only part of the dental arches crossed by X-rays up to that time

is generated. The image that comes out is the same when the X-ray tube stops against the back or shoulders of larger or kyphotic patients (Figure 9).

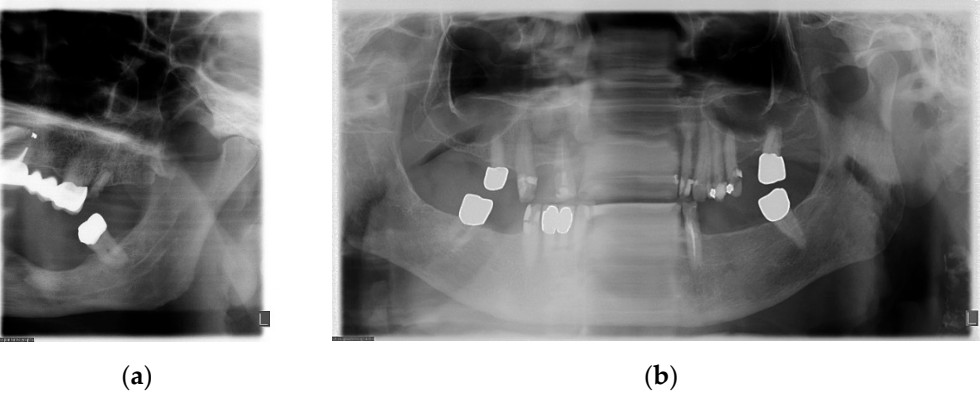

(**a**)                                                                 (**b**)

**Figure 9.** Incomplete acquisition: (**a**) X-ray tube halts against the patient's right shoulder during the initial phase of the tube rotation. Only the posterior structures of the left side are represented in the image. (**b**) Slowing down without stopping the X-ray tube because of patient's humpback. The initial and end phases of the tube rotation do not slow down with consequent normal visualization of the left and right posterior areas, respectively.

## 7. Metal and Motion Artefacts

Several objects result in the generation of metal-related artefacts, including the presence of metallic materials, dental crowns, wires, containment plates, earrings, necklaces, and machine parts [16]. Metal objects hinder the visualization of the anatomical structures where superimposition occurs. Moreover, the technical principles of image acquisition based on curved rotational zonography cause the projection of objects to become deformed images on the contralateral side (Figures 3 and 10). To avoid metal artefacts, it is only right to remove all removable metallic materials from the head and neck area, including mobile prostheses [16].

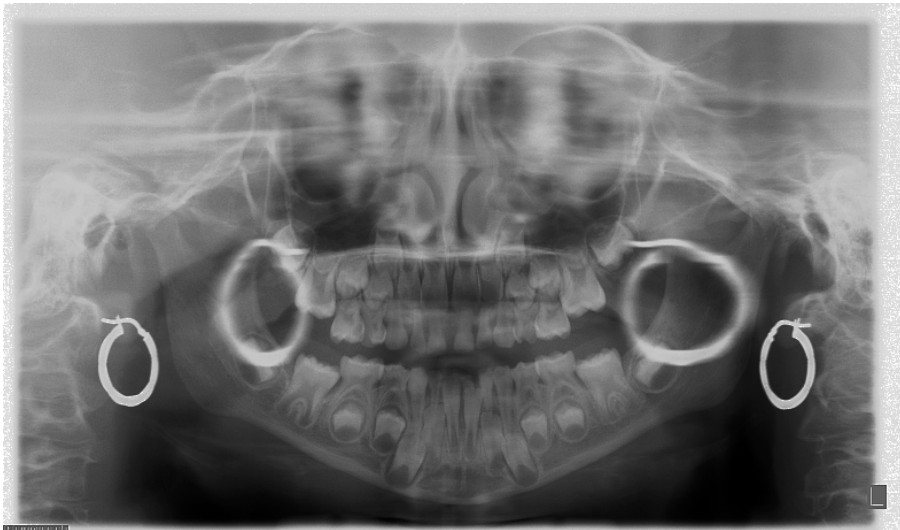

**Figure 10.** Artefacts by earrings. Note that the shape of each earring is mirror-like and is reflected on the contralateral mandibular ramus/third molar area in the form of an artefact.

PAN acquisition time is short, around 12–15 s. However, slight patient movements may occur, causing motion artefacts such as blurring or double-edged effects (Figures 11 and 12). These are most frequently found in children and elderly patients due to their limited capability to maintain the correct position for the required time [17,35,37].

Patient cooperation is thus fundamental to achieve adequate image quality. In cases of reduced patient compliance, the application of a fast scanning mode may be considered [17,37].

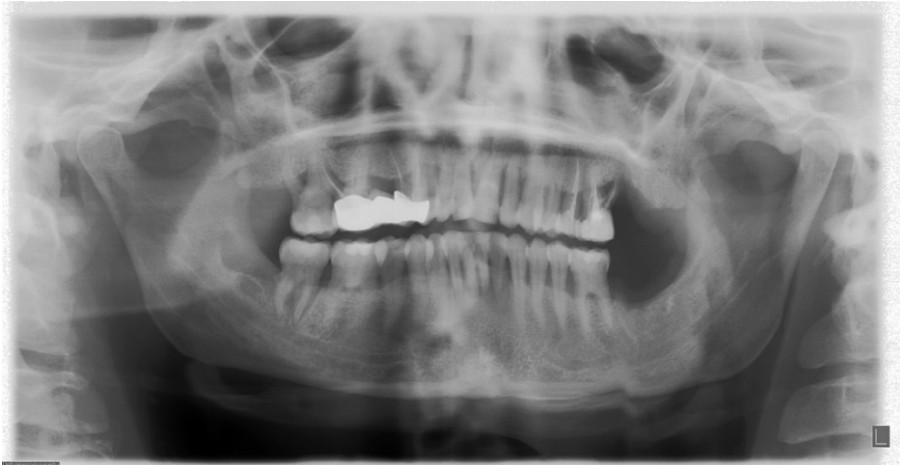

**Figure 11.** Motion artefact represented by the stepped-stairs effect on the mandibular lower edge below the left first molar due to the rapid (abrupt) movement of the patient before X-ray tube crosses over their back.

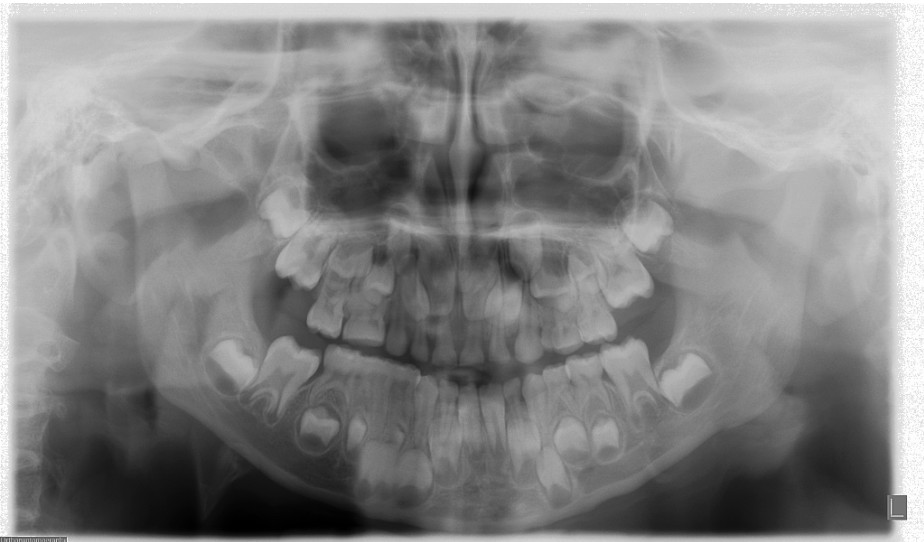

**Figure 12.** Motion artefact. The movement of a young patient after X-ray tube crosses over their back can be observed as a blurred and distorted image on the mandibular canine/premolar area on the right side.

## 8. Newest Advances for PAN in a Digital Era

Recently, advances in PAN have had a significant impact on improving image sharpness without neglecting any aspects related to patient health. The design of state-of-the-art dental panoramic units is based on multi-layer recording, flexible radiation beam collimation, quick scan modes, the correction of positioning errors and motion artefacts, and reductions in effective doses [27].

Multiple panoramic imaging generates a set of several parallel and thin layers within the focal trough so that manual or automatic image postprocessing enables one to choose the best segments of one-time registered layers and gather them in a high-quality PAN. This system makes it possible both to more accurately examine patients with malocclusion

or maxillofacial asymmetries and to correct positioning errors on the anteroposterior plane due to the incorrect location of incisors on the bite block [27].

Different radiation beam collimations allow one to acquire different kinds of PAN, including the orthogonal PAN that reduces crown overlapping of adjacent teeth, which provides a better periodontal analysis; selectable segmentation into two (hemi-panoramic) or four quadrants with optimized projections to reduce the radiation dose; and four-segment bitewing exposures that are limited to crowns, though this is an alternative technique to intraoral imaging and is prized by patients with strong gag reflexes when it comes to examining interproximal caries. In addition, specific protocols for pediatric patients are capable of limiting radiation doses by means of a reduction in exposure parameters and times. This is made possible because children have small mouths; therefore, the robotic C-arm in panoramic machines can carry out a rotation arch that is shorter than normal by removing the initial and final parts of the exposure [17,37]. Some dental panoramic units automatically adjust the size of the field of view in relation to a patient's date of birth, while others implement technologies that recognize the individual dental arch morphology associated with the best focusing selection systems to optimize exposure and scan times for people of all ages. Modern machines may be also equipped with a quick scan mode. Regardless of exposure time, this is a high-speed scan option that enables the performance of a PAN in 7 to 10 s. The short scan time reduces the chance of movement in uncooperative patients such as small children, old adults, and people with motor disabilities. When associated with low exposure times, a fast PAN significantly reduces the delivered radiation dose; therefore, such scans should be used in children for primary investigations and in cases of multiple follow-ups [27]. Furthermore, algorithms compensating for a patient's movement may be available, making it possible to correct a limited range of movement (up to 1.5 cm). This option is not only useful for children but also for patients with involuntary movements, e.g., Parkinson's disease.

Some dental panoramic units are fitted with cameras that enable the verification of both the choice of shape/size of dental arches that come closest to those of patients and their positioning by different colored lines corresponding to acceptable or incorrect localizations of horizontal and vertical reference planes. An image preview may be also visible live on the screen panel during the examination to verify potential patient movement. Such imaging helps us to understand whether definite movements may cause artefacts in the final image and whether it is necessary to stop the robotic C-arm before the exposure ends, since PAN will not accept it diagnostically and therefore has to repeat the exposure [19].

Finally, PAN images can originate from both sensors for panoramic X-rays only and combined sensors for CBCT and panoramic X-rays. Such combined sensors within the same dental unit are becoming more common in clinical practice, and they have given rise to so-called "hybrid machines" that combine PAN with a relatively small to medium field of view CBCT system. Additional programs in software packages may also generate two-dimensional image samples—including PAN—from a volume acquired with low-dose CBCT scans [27].

## 9. Conclusions

The performance of a suitable PAN depends on several parameters. Execution errors significantly affect image quality and most of them are related to patient positioning and adequate instructions. A systematic approach to PAN performance optimization is recommended to improve overall examination quality.

**Author Contributions:** Conceptualization, R.I. and C.N.; methodology, M.N. and G.A.; validation, G.A., L.C. and C.N.; investigation, L.C.; resources, C.N.; data curation, L.C. and G.A; writing—original draft preparation, R.I., F.G. and C.N.; writing—review and editing, R.I., F.G. and C.N.; visualization, M.N. and L.C.; supervision, F.G. and C.N.; project administration, R.I. and C.N. All authors have read and agreed to the published version of the manuscript.

**Funding:** This research received no external funding.

**Institutional Review Board Statement:** Not applicable.

**Informed Consent Statement:** Informed consent was obtained from all subjects involved in the study.

**Data Availability Statement:** Not applicable.

**Conflicts of Interest:** The authors declare no conflict of interest.

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
