# Peer review of "Basic Knowledge and New Advances in Panoramic Radiography Imaging Techniques: A Narrative Review on What Dentists and Radiologists Should Know"

_applsci, doi:10.3390/app11177858_

Round 1

Reviewer 1 Report

Dear authors,

It is a pleasure to be a reviewer of your publication. I found it very nicely and clearly prepared. It is true that OPG is very important diagnostic tool in dental surgery especially in the first stages of diagnosis and evaluation of treatment effects. In my opinion the publication is valuable and deserved to be published, as there are no studies summarizing the advantages and disadvantages of OPG and the most common errors related to this diagnostic tool.

My minor comments are:

  1. The title may be more informative if contains information about the type of publication, e.g. narrative review.
  2. Is it possible to find data about the most frequently difficulties and errors in taking OPGs and their evaluation?

Best regards

Reviewer 2 Report

Overall, the manuscript is well-written and providing details on accurate image recording appears relevant.

However, the article is very general and does not provide new insights compared to book articles written in the 1990s.

In recent years, many new features were developed for panoramic xrays, including multi-layer recording, opportunities to correct the images if the head is too much nodded down, etc.
in my opinion, these novel features need to be mentioned in the article.

additionally, options for collimation and dose reduction, such as "quick optg" have to be mentioned.

in the introduction, disadvantages of panoramic xrays are summarized. however, the advantage is the better number of line pairs compared to cbct;

please mention typical doses and differences in sensors (combined sensors for cbct/ panoramic xrays), sensors for panoramic xrays only, digital versus analog technology;

eventually, please consider that the abbreviation OPG is colloquial, it should be OPTG or OPT, since the term is OrthoPanTomogram

Round 2

Reviewer 2 Report

All my concerns have been resolved